# Deformation Analysis of Different Lithium Battery Designs Using the DIC Technique

**Szabolcs Kocsis Szürke** [ID], **Mátyás Szabó, Szabolcs Szalai and Szabolcs Fischer** * [ID]

Central Campus Győr, Széchenyi István University, H-9026 Győr, Hungary; kocsis.szabolcs@ga.sze.hu (S.K.S.); szabo.matyas@sze.hu (M.S.); szalaisz@sze.hu (S.S.)

*   Correspondence: fischersz@sze.hu; Tel.: +36-(96)-503-400

**Abstract:** The growing number of electric vehicles and devices drives the demand for lithium-ion batteries. The purpose of the batteries used in electric vehicles and applications is primarily to preserve the cells and extend their lifetime, but they will wear out over time, even under ideal conditions. Most battery system failures are caused by a few cells, but the entire system may have to be scrapped in such cases. To address this issue, the goal is to create a concept that will extend the life of batteries while reducing the industrial and chemical waste generated by batteries. Secondary use can increase battery utilization and extend battery life. However, processing a large number of used battery cells at an industrial level is a significant challenge for both manufacturers and users. The different battery sizes and compositions used by various manufacturers of electric vehicles and electronic devices make it extremely difficult to solve the processing problem at the system level. The purpose of this study is to look into non-destructive battery diagnostic options. During the tests, the condition of the cells is assessed using a new diagnostic technique, 3D surface digitalization, and the fusion of electrical parameters. In the case of surface digitalization, the digital image correlation (DIC) technique was used to estimate the cell state. The tests were conducted on various cells with widely used geometries and encapsulations. These included a lithium polymer (soft casing), 18650 standard sizes (hard casing), and prismatic cells (semi-hard). The study also included testing each battery at various charge states during charging and discharging. The findings help to clarify the changes in battery cell geometry and their localization. The findings can be applied to cell diagnostic applications such as recycling, quality assurance, and vehicle diagnostics.

**Keywords:** lithium battery; battery deformation; battery testing; DIC; SOC; reusability

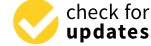



## 1. Introduction

Over the last decade, global carbon dioxide emissions from fossil fuels have steadily increased, with the transportation sector accounting for more than 20% [1–3]. Various laws are being implemented in the European Union to reduce these emissions and encourage the use of electric vehicles [4,5]. Relevant transportation issues, such as improving energy efficiency [6–9], lowering local carbon dioxide emissions, and reducing noise levels, have focused on researching and developing battery technologies [10,11]. The European Union's proposal and acceptance of the 'Fit for 55' legislative package, which includes regulations aimed at achieving a 55% reduction in emissions by 2030, has fueled interest in batteries even further [12].

Lithium-based battery technology is one of the most efficient and widely used in batteries, with applications ranging from automotive to entertainment electronics to space exploration. Their popularity stems from their high energy density, long lifespan, and availability in various forms, allowing them to be easily implemented as energy storage solutions. They have a low self-discharge rate, but their main disadvantage is that they are significantly more expensive to manufacture than other battery technologies (e.g., the lead-acid battery, or NiCd battery) [13]. Lithium batteries are typically closed systems

with variable geometries, including a positive electrode (cathode), a negative electrode (anode), separators, and electrolytes. Electric and hybrid vehicle energy storage systems are built by connecting hundreds of these battery cells in series and parallel, along with protective electronics and packaging. As a result, these systems have evolved into highly complex, safety-critical units that necessitate continuous monitoring and diagnostics. When building battery systems with lithium-ion (Li-ion) cells, various issues can arise, including overcharging and deep discharge, resulting in high temperatures, gas generation, and, in worst cases, thermal runaway [14]. Thermal abuse in Li-ion batteries is caused by overheating due to mechanical or electrical abuse, as well as improper connections. Thermal runaway (TR) causes an increase in internal and external temperatures, frequently resulting in explosions. TR mechanisms are studied using various methods, including extended volume-accelerating rate calorimetry tests [15]. Battery Management Systems (BMS) are critical for ensuring the safety of such energy storage systems in order to avoid such problems [16]. Furthermore, they play an essential role in battery thermal management, State of Charge (SoC), State of Health (SoH), and Depth of Discharge (DoD) estimations, providing critical information about a cell's or module's current state [17]. Refer to the following articles for more information on BMS reliability: [18,19]. Control fault modes include battery and sensor failures, as well as operational (BMS) failures. Overcharging, overloading, deep discharge, overheating, external/internal short-circuiting (ESC, ISC), electrolyte leakage, battery deformation, rapid battery degradation, and thermal runaway are some of the battery faults mentioned previously. The majority of these faults are caused by battery aging.

## 1.1. The Most Common Li-Ion Battery Geometries

The geometric design of batteries is one of the most important factors to consider when installing them in vehicles. Today's most commonly used forms in the automotive industry are cylindrical and prismatic, with soft and hard casing variants [13]. Because of their smaller and more standardized sizes, prismatic cells are primarily used in the automotive industry, whereas cylindrical designs find more general applications. The most common size is the 18650 cell [20], widely used in vehicles and flat packs, smaller and larger energy storage systems (power banks and power walls), hand tools, flashlights, and other applications. Because of the growing popularity of electric vehicles and portable electronics, research on battery diagnostics and testing is becoming more widely available [21]. Table 1 summarizes the benefits and drawbacks of batteries with various cell geometries [22].

**Table 1.** Advantages and disadvantages of prismatic, cylindrical, and pouch cell geometries.

| Battery Type | Advantages | Disadvantages |
|---|---|---|
| Cylindrical cell | Cells and battery modules can be easily manufactured automatically. | Using cylindrical cells in a stack or module leads to lower energy density (the cell's circular cross-section does not allow maximum use of the available space). |
| | The cylindrical shape creates more space between the cells, allowing optimal heat management. | The large number of cylindrical cells requires many contact arrangements, which increases the complexity of battery–battery assembly. |
| | The circular shape of the battery provides high mechanical stability, as the internal pressure from lateral reactions is evenly distributed. | |
| Prismatic cell | Optimal use of space in the battery pack. | The electrode and separator sheets at the container edges have a higher resistance against stress. |
| | The deformation of prismatic cells is not significant. | |
| | Higher nominal capacity (Ah) and energy density can generally be achieved. | Prismatic cells can typically be more expensive to manufacture than other cell types. |
| Pouch cell | It has the lowest weight of all these shapes. | Most pouch-type cells need to be individually customized, which increases manufacturing costs. |
| | Flexible design (complex shapes can be created). | Extra protection against external influences must be provided. |

### 1.2. Cell Deformation Processes and Investigation Possibilities

Under certain operational conditions, changes in the chemical composition of the active materials in batteries can result in irreversible volume changes due to the exponential increase in charge–discharge cycles. These conditions can cause significant electrode deformation and degradation and increased impedance [23]. The irreversible volume change is caused by the electrodes' reaction with the electrolytes, which results in their insoluble deposition on the electrode surface. As a result of the continuously increasing cycle number, various mechanical reactions occur in various Li-ion cells, resulting in capacity losses and various failures [24]. Lithium diffusion can be seen in and out of the electrodes during the charging and discharging of Li-ion batteries. If the charge/discharge rate is slow and there are no other external constraints to uniform diffusion, the distribution of lithium in the particles is nearly homogeneous, allowing the particles to expand and contract without stress. However, contrary to theoretical predictions, the distribution of lithium is inhomogeneous, resulting in particle stress and subsequent electrode damage or fracture [25]. Current cathodes experience significant volume changes, which cause capacity fading, microcracks, and fractures. Xinye Zhao et al. address volume changes in cathode materials during lithium-ion battery cycling, emphasizing the importance of developing zero-strain (zs) cathodes to mitigate issues such as capacity fading, microcracks, and fractures caused by significant volume changes [26]. Thermal heat release in batteries can be caused by thermal (overheating), electrical (e.g., overcharging/discharging or significant transient current loads), or mechanical (fracture, impact) stresses [27]. An uncontrolled and uncontrollable increase in temperature occurs in a cell during thermal heat release. This can produce a variety of gases, increasing the internal pressure [28]. The overcharging mechanism, for example, has two significant effects. First, lithium dendrites (LDs) form due to the extreme lithium intercalation at the anode surface. Second, severe lithium de-intercalation causes the cathode structure to collapse, resulting in heat and oxygen leakage. The release of oxygen accelerates electrolyte breakdown, resulting in significant gas emissions. Increased internal pressure can cause venting, which increases heat output due to the interaction of air and reacting materials inside the cell [15]. The increased internal pressure can cause the deformation of the battery and its housing components. In summary, battery cells can deform due to external mechanical impacts [29], most notably over-discharging, overcharging, and high cycle numbers caused by general use [30,31]. The reaction of the electrodes with the electrolytes, the development of inhomogeneous particle distribution during lithium diffusion, and the "natural" aging due to high cycle numbers can all be triggering factors [32]. The thermal processes and the chemical processes they induce can cause significant deformation in pouch cells in most cases [33,34]. Numerous studies are focusing on integrating machine learning (ML) techniques to improve battery thermal and thermal management (BTM) systems for lithium-ion batteries, particularly in electric vehicles, to reduce thermal processes and avoid thermal runaway. It emphasizes the significance of creating effective BTMs to meet lithium-ion batteries' stringent temperature requirements. This study delves into various machine learning (ML) models, such as artificial neural networks (ANN), convolutional neural networks (CNN), and long short-term memory (LSTM), highlighting their applications in optimizing BTMs, predicting heat generation and temperature, and their overall system optimization [35,36].

Because of their deformation and bending, which cannot be controlled by voltage and temperature, lithium-ion battery safety research remains a priority in the engineering and scientific communities [37]. As a result, understanding the mechanical properties and failure mechanisms is becoming increasingly important. Cost-effective investigations for cells without hard casings (e.g., pouch cells) can include observing deformation with tactile or strain sensors [38]. Deformation is frequently investigated in measurements using multi-point distance sensors [39]. Casings for commercially available 18650 cells are typically made of steel with a thickness of 0.23–0.3 mm [40]. Several studies [37,41] investigate 18650 cells under various mechanical stress tests. These mechanical tests, supplemented by finite element simulations, monitor cell behavior based on different compression angles and

contacts, considering temperature and voltage. According to Spielbauer et al., a 1 mm deformation of the cell housing can be fatal [42]. During compression tests, Szabo et al. attempted to detect short circuits [41]. They used compression tests and finite element modeling to validate the experimental results with deformations ranging from 0 to 7.5 mm. Wenfeng Hao et al. used 3D DIC, an infrared camera, an ultra-depth microscope, and acoustic emission (AE) to investigate the mechanical properties of 18650 cells during compression [37]. They discovered that as the SoC increased, the bending modulus and stiffness of the 18650 cells increased, as did the delamination, interlayer slip, and electrode breakage within the cells. Several studies [43–45] discuss the computer tomography (CT) examination of battery cells. Yikun Wu et al. used CT scans to observe changes in the roundness of the casing in 18650 Li-ion cells [43]. The 18650 Lishen cell under consideration had a nominal capacity of 2.7 Ah, a silicon-graphite composite anode, and an NMC (Lithium Nickel Manganese Cobalt Oxide) cathode. They created CT images during the experiment using static strain gauge cells, an infrared camera, and an accumulator testing system. They discovered that changes in the roundness of the steel casing were caused by thermal expansion and the heat generated by chemical reactions during the early cycling stages. They measured inhomogeneous and negative deformations in the casing after prolonged cycling (>300).

When researching batteries, one must consider the effects of commercial cathode materials and the performance of each cell design. Although lithium ferrophosphate (LFP) cathodes have excellent thermal stability and safety, they frequently have lower energy density than other cathode materials, which affects the overall energy storage capacity and range of the battery cells. Due to their reactive nature, NMC cathodes have a higher energy density and better performance over a wider temperature range, but they can suffer from significant capacity fading and safety issues. While the cathode material used, such as LFP or NMC, has a significant impact on the energy, power, and safety of a battery cell, it also raises environmental and sourcing concerns, as the mining and processing of rare metals like cobalt and nickel can cause environmental harm and ethical concerns. The study by Sanad et al. [46] is critical because it proposes a viable method for improving the electrochemical performance of NMC811 cathode materials, which are commonly used in Li-ion battery technology, by coating them with a $ZnSnO_3$ perovskite film, potentially leading to more efficient and long-lasting batteries. NMC cathode materials, particularly NMC811, exhibit significant difficulties, such as rapid capacity fading and structural damage due to side interactions with acidic electrolyte species during charge/discharge processes. By coating these materials with a $ZnSnO_3$ perovskite film, the electrochemical performance of these materials can be significantly improved, resulting in increased thermal stability, better cycle stability, higher capacity retention, and fewer side reactions with the electrolyte.

Ran Tao et al. used scanning electron microscopy and digital image correlation to quantify 2D anisotropic displacement and strain fields in graphite-based electrodes [47]. They discovered a 50% irreversible expansion in the first cycle, followed by reversible anisotropic deformation in the second cycle, depending on the spatial directions. The expansion and deformation were more pronounced at the electrode-separator interfaces, where lithiation and delithiation occurred [47]. Weiping Diao et al. used a CT and an SEM (Scanning Electron Microscope) to study the deformation of windings and the causes of thermal heat release in cells in various states of charge [48]. Their most important discovery was that the occurrence of electrode layer fractures was determined by the SOC level of the charge threshold rather than the charging C-rate and cell temperature [48]. Furthermore, electrochemical impedance spectroscopy (EIS) can be a valuable tool for diagnosing the state of lithium-ion batteries and comprehending their aging mechanisms. EIS is used to investigate impedance changes, providing insights into the various frequency regions affected by degradation processes. This includes the high-frequency regions related to electrolyte resistance and the mid-frequency regions related to SEI layer growth and charge transfer processes [49]. Due to the wear and aging of current systems, reuse and recycling approaches are becoming increasingly important. On the other hand, non-destructive

cell examination methods are required for reusability. The most commonly used capacity tests currently take several hours, and inspecting each cell in a vehicle can take days or weeks. However, the examination time becomes one of the most critical factors for many energy storage systems. Finding the quickest and most accurate approach during condition monitoring is therefore critical. Several studies have focused on various diagnostic tests that use structural changes to determine battery condition [50]. Since there is often a correlation between a cell's external deformation and its State of Health (SoH), there has been an increased demand for diagnostic research related to the deformation of individual battery cells in recent times. Because of the growing number of electric vehicles and the recyclability of aging batteries, a relatively quick and reliable diagnostic system that can easily identify the state of a given cell without direct destruction is required [51]. There has been little research into identifying critical areas for deformation. The primary goal of the research presented here is the non-destructive diagnostic examination of Li-ion batteries with different geometries. Through the established non-destructive diagnostic system, it is possible to determine potentially deforming areas more accurately during the adaptation of batteries for the automotive industry. The diagnostic system is based on a high–precision contactless optical measurement technique called DIC. The surface digitalization uses the GOM Metrology ATOS measuring system on several Li-ion battery cells with different SoH and designs. The observation of deformation takes place during various charge–discharge cycles.

## 2. Materials and Methods

The increasingly popular 3D DIC technology has recently enabled the contactless measurement of the displacements and deformations of objects subjected to mechanical or environmental loads. The main advantage of this system is its ability to provide highly accurate three-dimensional measurement data, even when used in an industrial setting, thereby assisting with quality assurance in manufacturing processes. The contactless, full-space measurement of the displacements and deformations of an object subjected to mechanical, thermal, or environmental loads is possible with 3D DIC technology. For image capture, stereo cameras are required, as is a computer system for synchronization, data storage, and processing [52]. Cell deformations were measured in the current study using the GOM ATOS Triple Scan (Figure 1—part (a); GOM GmbH, Braunschweig, Germany), a high-precision 3D industrial scanner [53].

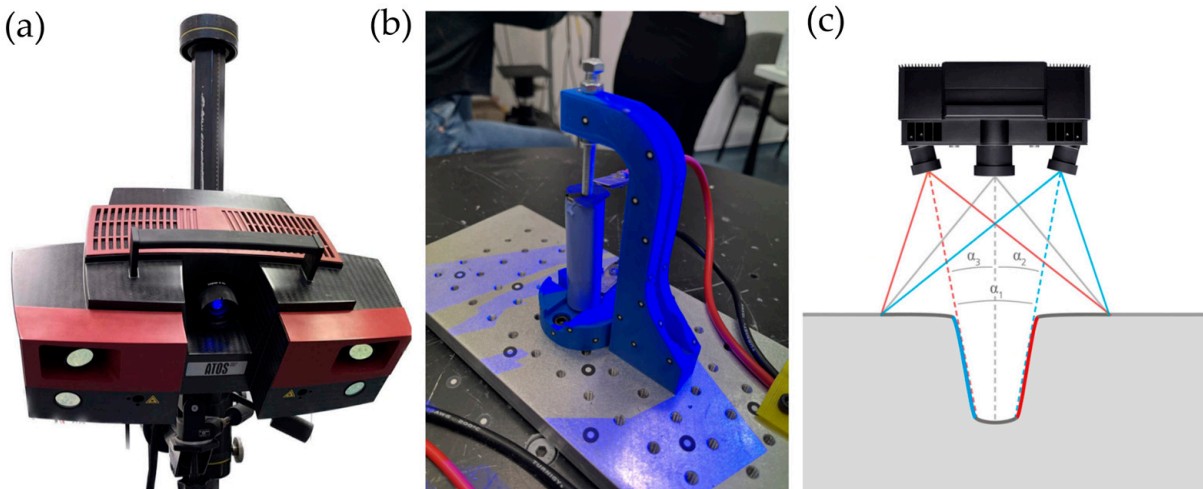

**Figure 1.** (**a**) The ATOS Triple Scan; (**b**) optical scanner measurement volume determination; (**c**) 18650 cell measurement process.

The system's cameras and projector are calibrated, so the three devices can use the triangulation method to determine points on a 3D surface from any two viewpoints

(Figure 1—part (b)). The system can measure small objects with micron precision. Larger objects can be inspected within a few tenths of a millimeter precision. This is determined by the type of measurement lens used. A single image cannot usually capture the entire object under study, and the cells used in this study necessitated multiple digitalization angles. As shown in Figure 1—part (c), reference points are used to connect the various images. For the GOM software (version 2019), points are used to convert images taken from various perspectives into a single point cloud [38]. In these tests, in addition to the DIC parameters, the electrical parameters had to be determined in order to evaluate the measurement. The saved parameters during battery measurement are the cell voltage, load, and charge current, and the derived quantities are the capacity, SoC, and SoH. A HAMEG HMP 4030 3-channel 384 W power supply (München, Germany) is used for charging, an EA-EL 3160-60 400 W dummy load (Viersen, Germany) is used for discharging, and a National Instruments USB-6341 (Austin, TX, USA) is used for data acquisition. A control program written in LabVIEW (Version 14.0.1f11) ensures safe operation.

Charge and discharge measurements were taken in all cases. In addition to the deformation, the current, voltage, and interruption time were recorded for analysis, and the SoC level was calculated from these values. Several interruptions were made during charging and discharging to monitor the change. A highlighted measurement during the cylindrical battery measurements is shown in Figure 2.

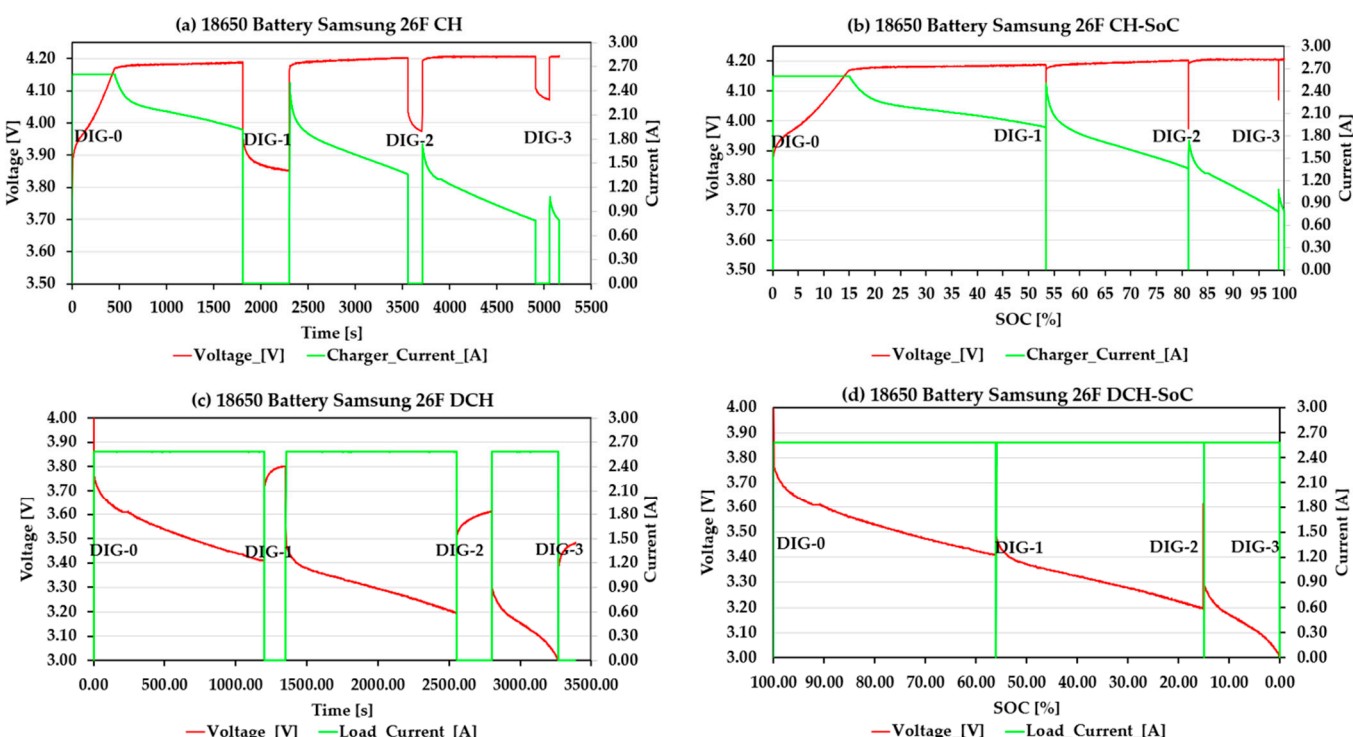

**Figure 2.** (**a**) Charge profile of the battery; (**b**) Charge profile of the battery as a function of the SoC; (**c**) Discharge profile of the battery; (**d**) Discharge profile of the battery as a function of the SoC.

The charging and discharging currents (in green) and the sampled voltage (in red) are shown in Figure 2. The current and voltage sampled during charging are shown in Figure 2a, and the exact measurement is a function of the SoC in Figure 2b. Figure 2c shows the discharge test results as a function of time, and Figure 2d shows the results as a function of the SoC. The figures show several interruptions and digitalizations were performed, with three deformation states recorded in both cases. The "0" or initial state was recorded before the discharge or charge began, and this was used to create the CAD model, which was then compared to the other states. The remaining states, 1-2-3, all indicate a break in the digitalization process. It is worth noting that, on average, more than ten interruptions

were made in most cases. Although the number of these was not repeated simultaneously, the SoC level was calculated in all cases. The general digitalization process was as follows:

- Fully charge the battery to achieve a 100% SOC.
- Surface digitalization of the fully charged state. Create the initial CAD model.
- Start loading and digitizing after each interruption.
- Determination of the deviation of the deformation from the initial state. By default, a discharge during a contraction is the expected presence, which subsequently appears in the data—as a sign.
- Calculate the energy extracted during the interruption using the coulomb counting method. Generation of the SoC for the deformation data.
- Achieving a fully discharged state.
- Mapping separate interrupt data and SoC values into a matrix.
- One hour of rest.
- Surface digitalization of the fully discharged state again. Create a starting CAD model.
- Determination of the deviation of the substitution from the initial state. By default, the swelling during charging is the expected presence, which then appears as a positive value in the data.
- Calculate the energy input in the interrupt snapshot using the coulomb counting method. Generation of the SoC for the deformation data.
- Reaching a fully charged state.
- Mapping the separate interrupt data and SoC values into a matrix.
- Evaluation of the results.

Different batteries were subjected to varying load and charge currents, but 1C was a guideline in all cases. The maximum voltage level was 4.2 V, with a minimum of 3 V (except for the lithium polymer battery, which was tested at a lower level).

In general, the following are the main steps in the measurement process: 1. connecting the battery to the test system (Figure 1—part (c)); 2. defaulting the cell's SOC value and digitizing. As a result, the reference model is being prepared for comparison; 3. beginning the charging and discharging process and interrupting it at regular intervals during specified periods; 4. performing one or more digitalizations during the interrupt time; and 5. comparing the measurement results to the initial state.

The tests aim to develop a battery testing procedure capable of classifying and grouping batteries in various states of degradation. The inspection system was expected to: 1. be capable of measuring cells of various types, capacities, and types of cells; 2. be capable of replacing batteries easily, quickly, and safely; 3. allow a wide range of battery deformation measurements; and 4. allow the condition of the cell to be determined with the most remarkable accuracy possible. Figure 3 depicts a measurement model of various cell shapes.

The cylindrical 18650-size cell is shown in Figure 3a, the prismatic cell is shown in Figure 3b, and the pouch cell is shown in Figure 3c.

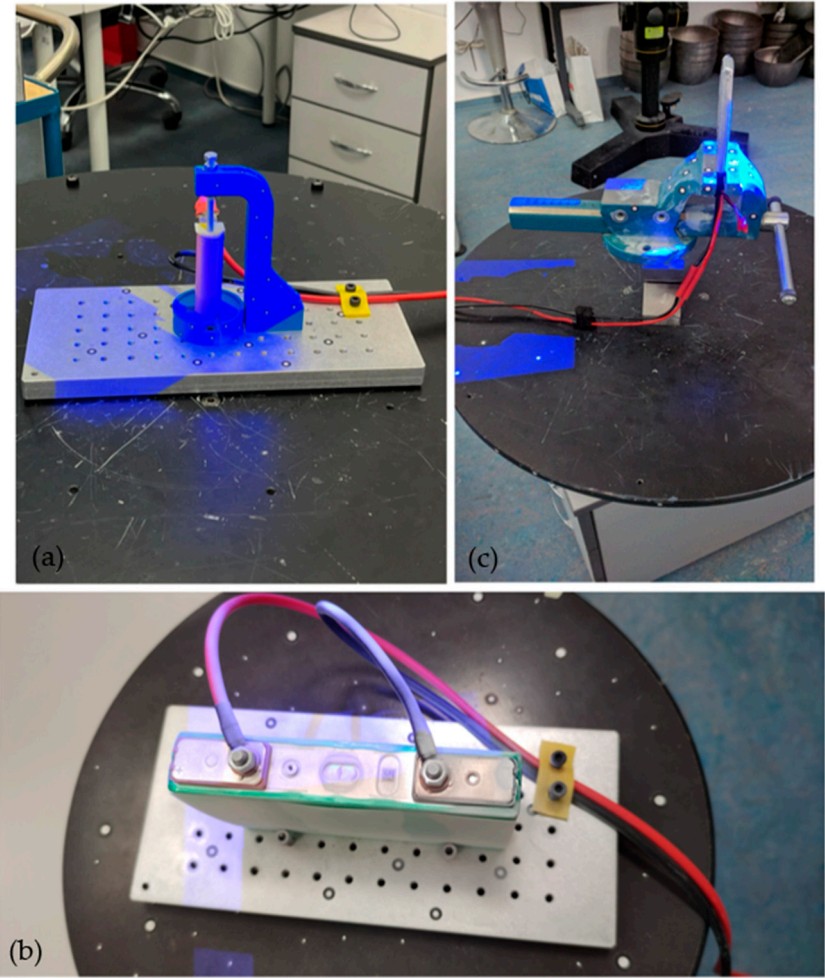

**Figure 3.** (**a**) Cylindrical cell mounting and testing; (**b**) Prismatic cell mounting and testing; (**c**) Pouch cell mounting and testing.

## 3. Results

The results of various battery design measurements are presented in separate subsections: the cylindrical-shaped battery analysis (Section 3.1), Prismatic-shaped battery analysis (Section 3.2), and pouch-shaped battery analysis (Section 3.3). In each case, the measurement procedure was the same (see Section 2 for a more detailed description), with DIC measurements taken at the point of interruption. After that, the electrical and digitalization data were fuzzed. The number of digitalizations varies, but in all cases, the results are presented as a function of the SoC and, thus, do not cause differences in comparability. Section 2 goes into greater detail about the instruments used and the measurement environment.

### 3.1. Cylindrical-Shaped Battery Analysis

This type of testing was done on 18650-size batteries. During the tests, newer and older batteries, as well as cells from various manufacturers, were all measured. The test results with the GOM measurement system are shown in Figure 4.

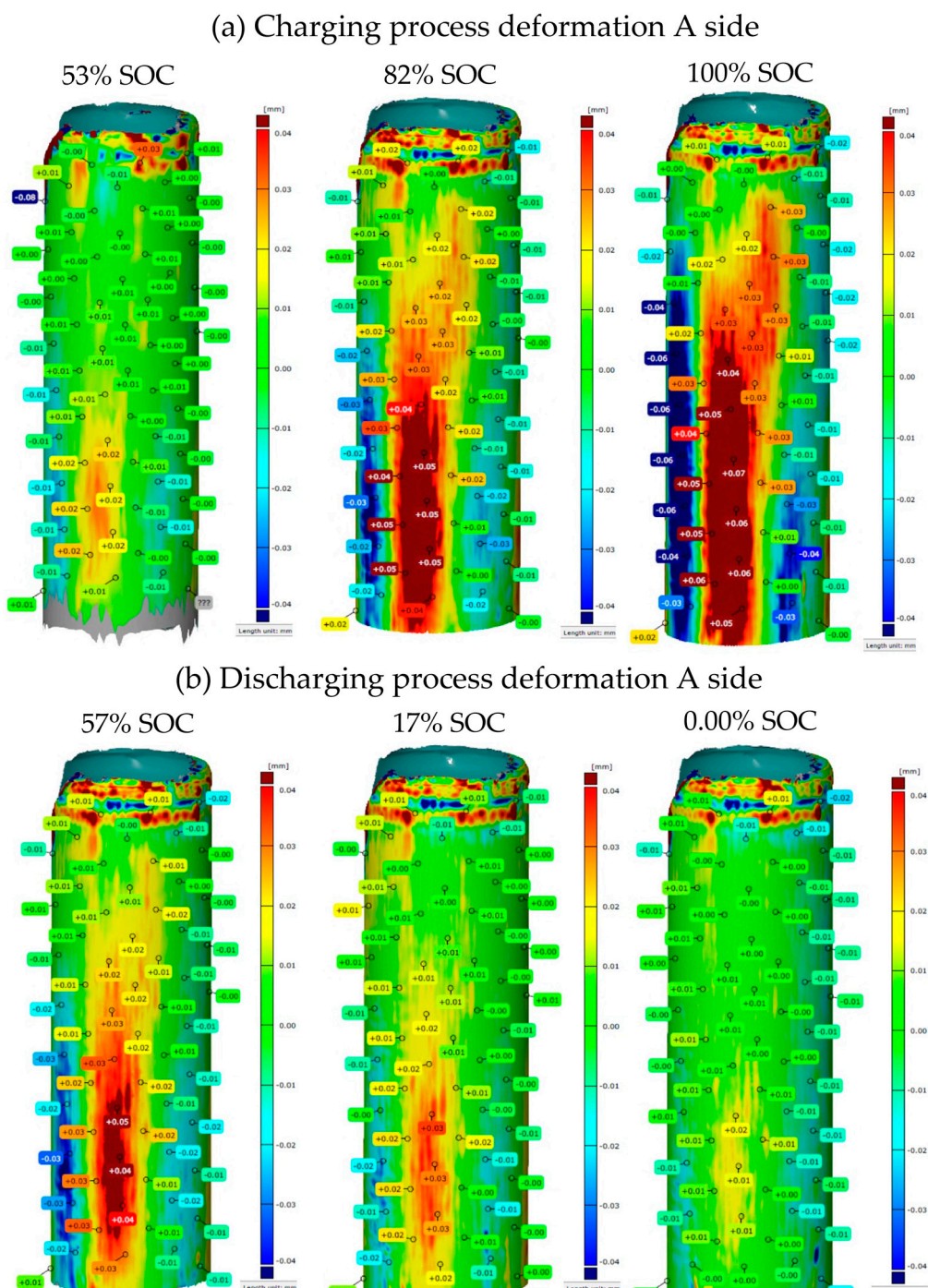

**Figure 4.** (**a**) 18650 battery deformation during charging; (**b**) 18650 battery deformation during discharge.

Figure 4a depicts the three changes that occur during charging, while Figure 4b depicts the results of discharging. It is important to note that the comparison results to the initial condition (CAD model) are presented in all cases. Millions of measurement points were taken, evaluated, and compared with the initial state by the GOM software during the digitalization process. However, this represents a large amount of data that would be difficult to use continuously (regarding the data interoperability and computational capacity). As a result, it is best to choose a few specific points that are evenly spaced. The GOM's software includes a meshing system that assists us in selecting a similar number of points with similar positions on a continuous basis. The next step in the evaluation is to

convert the cylindrical cell measurement data into a rectangular shape and organize the data into a matrix. The problem is that the measured cell has a regular cylindrical shape, making it difficult to determine an exact base point, especially for different cells. Thus, dividing 18650 cells into critical deformation zones is beneficial. The resolution approaches considered are depicted in Figure 5.

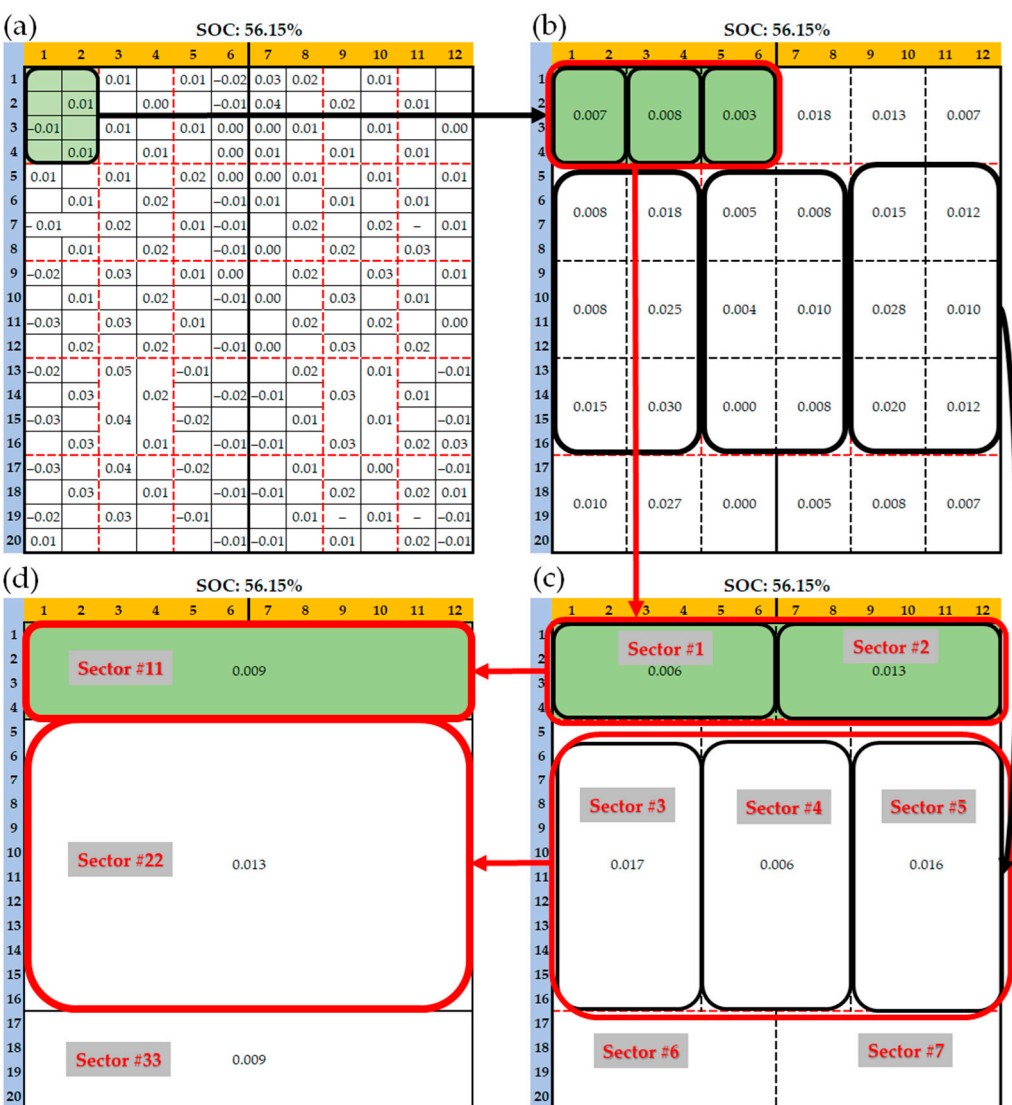

**Figure 5.** Assignment of critical zones in an 18650 cell of cylindrical design: (**a**) Values extracted from the GOM file at a constant distance; (**b**) 6 × 5 sector separation; (**c**) 7-sector separation; (**d**) 3-sector separation.

Figure 5 depicts the deformation data of the extended cylindrical cell. For the evaluation, a 1220 mesh was created (Figure 5a), which was later reduced to a 65 mesh (Figure 5b). Notably, in this matrix mesh, the average of the available values is always used; the amount of data is not constant. Because the first value in part (b) is the average of the sector highlighted in green in part (a), the 24 submatrix in Figure 5 has only one value. Part (b) is built at the end of the transformation from a total of 65 matrices that have been averaged over the previous state. Two approaches were used to process the data to determine the critical zone. The upper and lower parts of the cell were divided into two parts in the first case (Figure 5c), and the middle was divided into three zones. This allows the researchers to compare each zone to the others. The other approach divides the cell into three sections (Figure 5d). Similar to the process described above, the data were rescaled and organized

into larger matrices, with each larger matrix containing the average of the previous values. After that, the electrical and digitalization data were combined and fused. Table 2 shows the averaged deformation values measured during the charging and discharging of several 18650-size batteries. The values in the table are for Figure 5, part (a).

**Table 2.** Deformation of an 18650-size battery with a cylindrical design based on min/max values measured over the entire surface.

| SOC [%] | CH/DCH | Deformation [mm] | Variance Range [mm] | | Deviation [mm] |
|---|---|---|---|---|---|
| 0–25 | CH | 0.017 | 0.000 | 0.080 | 0.010 |
| | DCH | −0.052 | −0.060 | 0.060 | 0.020 |
| 25–50 | CH | 0.033 | 0.000 | 0.080 | 0.017 |
| | DCH | −0.033 | −0.044 | 0.050 | 0.005 |
| 50–75 | CH | 0.050 | 0.000 | 0.080 | 0.010 |
| | DCH | −0.025 | −0.030 | 0.050 | 0.004 |
| 75–100 | CH | 0.060 | 0.000 | 0.050 | 0.010 |
| | DCH | −0.014 | −0.060 | 0.060 | 0.000 |

The charge level values in Table 3's left column are in percentages, while the values in the other columns are in millimeters. In contrast to the previous analysis, seven distinct sectors were examined. As in the previous analysis, the SoC has been divided into four sectors. The extent of the deformation was smaller in each case in this approach, which is also due to the nature of the analysis, but the average value of each sector was used. There are also more considerable differences between sectors. The sector 4 values for discharge and sector 5 values for charging are the most useful for further analysis. The values of the different sectors are used and understood at the respective charge level to obtain the range of variation. Variance values are also calculated for each SoC level and region. The values in Table 4 are based on Figure 5, section (d).

In Table 4, the first column shows the SoC level; the other values are deformation-related and are in millimeters. The analysis procedure was very similar to the previous one, except for reducing the seven sectors to three (as shown in Figure 5). The average values were also used for the sector transformations. Range and multiplication analyses were also carried out in the same manner as previously described (Table 3). The analysis results show that as the deviations are averaged, they become smaller and smaller, almost within the measurement system's error range.

**Table 3.** Deformation of the cylindrical 18650 battery in the cases of 7 sectors.

| SOC [%] | CH/DCH | Sec_1 | Sec_2 | Sec_3 | Sec_4 | Sec_5 | Sec_6 | Sec_7 | Variance Range [mm] | | Deviation [mm] |
|---|---|---|---|---|---|---|---|---|---|---|---|
| 0–25 | CH | 0.012 | 0.008 | 0.008 | 0.006 | 0.012 | 0.010 | 0.007 | 0.006 | 0.012 | 0.002 |
| | DCH | 0.008 | 0.026 | 0.021 | −0.011 | 0.021 | −0.003 | −0.008 | −0.011 | 0.026 | 0.015 |
| 25–50 | CH | 0.012 | 0.009 | 0.012 | 0.006 | 0.015 | 0.013 | 0.011 | 0.006 | 0.015 | 0.003 |
| | DCH | 0.007 | 0.025 | 0.016 | −0.009 | 0.011 | −0.002 | −0.006 | −0.009 | 0.025 | 0.012 |
| 50–75 | CH | 0.021 | 0.011 | 0.030 | 0.012 | 0.018 | 0.028 | 0.012 | 0.011 | 0.030 | 0.008 |
| | DCH | 0.006 | 0.021 | 0.015 | −0.007 | 0.008 | −0.002 | −0.004 | −0.007 | 0.021 | 0.010 |
| 75–100 | CH | 0.030 | 0.016 | 0.036 | 0.016 | 0.015 | 0.033 | 0.016 | 0.015 | 0.036 | 0.010 |
| | DCH | 0.003 | 0.012 | 0.012 | 0.001 | 0.015 | 0.004 | 0.004 | 0.001 | 0.015 | 0.006 |

**Table 4.** Deformation of the cylindrical 18650 battery in the cases of 3 sectors.

| SOC [%] | CH/DCH | Sec_1 | Sec_2 | Sec_3 | Variance Range [mm] | | Deviation [mm] |
|---|---|---|---|---|---|---|---|
| 0–25 | CH | 0.001 | 0.001 | 0.001 | 0.001 | 0.001 | 0.000 |
| | DCH | 0.004 | 0.003 | 0.000 | 0.000 | 0.004 | 0.002 |
| 25–50 | CH | 0.002 | 0.002 | 0.002 | 0.002 | 0.002 | 0.000 |
| | DCH | 0.002 | 0.001 | 0.000 | 0.000 | 0.002 | 0.001 |
| 50–75 | CH | 0.004 | 0.005 | 0.005 | 0.004 | 0.005 | 0.001 |
| | DCH | 0.001 | 0.001 | 0.000 | 0.000 | 0.001 | 0.001 |
| 75–100 | CH | 0.005 | 0.010 | 0.010 | 0.005 | 0.010 | 0.003 |
| | DCH | 0.000 | 0.000 | 0.000 | 0.000 | 0.000 | 0.000 |

In the cases studied in this study, the deformation of cylindrical batteries, including 18650 cells, did not reach the order of a decimal. It should be noted that the deformation measurements were taken during normal operation (no overcharging or over-discharge). Furthermore, the intermediate deformation values revealed that the most significant change occurred during charging at 80% (SoC) and discharging at the final 20% (SoC). After removing the casing (foil), no significant changes were observed in the cells. Furthermore, due to the averaging effect, the deformation rate is significantly reduced during sector breakdowns. Based on these findings, the tactile method is difficult to implement even with a seven-sector resolution, but the maximum/minimum value search method is recommended. However, using the DIC technique is justified for detecting more significant shape variation. Outliers should also be monitored and averaged in this case (averaging is required to detect erroneous outliers), possibly using OWA (Ordered Weighted Averaging).

*3.2. Prismatic-Shaped Battery Analysis*

The NMC Panasonic 25 Ah batteries of this type were tested. The study included both newer and worn-out cells. Figure 6 depicts the results obtained using the GOM measurement system.

Figure 6 depicts the digitalization results obtained at various SoC levels. It is important to note that there were several interruptions; the figure highlights the three states corresponding to the same measurement during charge and discharge. Furthermore, only one side of the cell is shown; of course, the values from both sides were used to evaluate the results. Furthermore, deformation occurred at different rates and locations (amorphous and asymmetric) for this type of battery. Figures depict contraction in blue and expansion in red. Figure 7 depicts the evaluation process.

Figure 7 depicts an analysis of the extended prismatic cell's deformation data. A 514 mesh was prepared for the evaluation, which was then reduced into 633 sectors. It should be noted that this matrix mesh does not always contain the same amount of data; the average of the available values was always used. In addition, columns 1–7 contain the values for the cell's side A, and columns 8–14 contain the values for the cell's side B. Two approaches were used to process the data to define the critical zone. The cell's upper, lower, and central parts were divided into two parts in the first case (Figure 7, part (c)). This allows the researchers to compare and contrast each zone. The cell is divided into lower, middle, and upper parts in the second approach (Figure 7, part (d)), and the sides (A and B) are merged. After that, the electrical and digitalization data were combined. Table 5 shows the averaged deformation values of several NMC Panasonic 25 Ah batteries during charging and discharging. Figure 7, part (a) shows the values in a table.

(a) Charging process deformation B side

(b) Charging process deformation B side

53% SOC

57% SOC

82% SOC

17% SOC

100% SOC

0.00% SOC

**Figure 6.** (**a**) Prismatic battery deformation during charging; (**b**) Prismatic battery deformation during discharge.

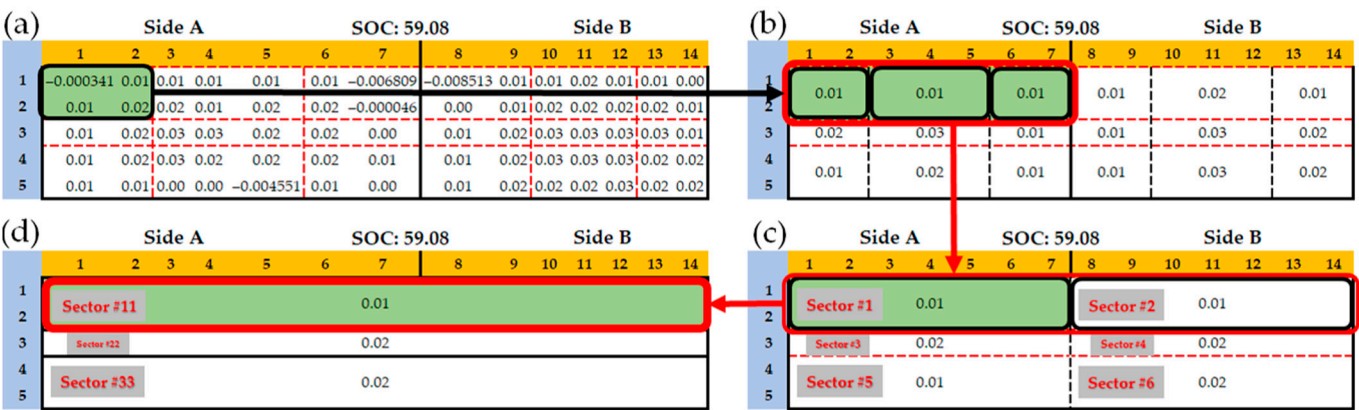

**Figure 7.** Assignment of critical zones in an NMC Panasonic 25 Ah cell of prismatic design: (**a**) Values extracted from the GOM file at a constant distance; (**b**) 6 × 3 sector separation; (**c**) 6-sector separation; (**d**) 3-sector separation.

**Table 5.** Deformation of Panasonic 25 Ah battery, an NMC composition-based battery with a prismatic design, based on min/max values measured over the entire surface.

| SOC [%] | CH/DCH | Deformation [mm] | Variance Range [mm] | | Deviation [mm] |
|---|---|---|---|---|---|
| 0–25 | CH | 0.022 | 0.000 | 0.038 | 0.016 |
| | DCH | −0.020 | −0.037 | 0.002 | 0.005 |
| 25–50 | CH | 0.032 | 0.013 | 0.038 | 0.004 |
| | DCH | −0.019 | −0.029 | 0.007 | 0.003 |
| 50–75 | CH | 0.043 | 0.017 | 0.050 | 0.002 |
| | DCH | −0.018 | −0.027 | 0.008 | 0.004 |
| 75–100 | CH | 0.049 | 0.016 | 0.055 | 0.003 |
| | DCH | −0.016 | −0.030 | 0.008 | 0.011 |

The charge level values in Table 5's first row are in percentages, while the values in columns 3, 4, and 5 are in millimeters. Section 3.1 (based on Table 2) describes in detail how the values in Table 5 are processed. A smaller data grid was used in this case after digitalization and subsequent comparison with the CAD model (measurement point selection). For each SoC level, five minimum (discharge) and five maximum (charge) values were used in the table and calculation. According to the results, the deformation rate is in the order of a hundredth. During charging, there is a slight continuous expansion, while during discharging, there is a slight continuous contraction. In this case, the deformation is less than that measured for the cylinder cells. Table 6 is based on Figure 7, section (c).

**Table 6.** Deformation of a Panasonic 25 Ah battery with prismatic NMC composition in the cases of 6 sectors.

| SOC [%] | CH/DCH | Sec_1 | Sec_2 | Sec_3 | Sec_4 | Sec_5 | Sec_6 | Variance Range [mm] | | Deviation [mm] |
|---|---|---|---|---|---|---|---|---|---|---|
| 0–25 | CH | 0.006 | 0.005 | 0.010 | 0.009 | 0.006 | 0.010 | 0.005 | 0.010 | 0.002 |
| | DCH | −0.011 | −0.008 | 0.010 | 0.001 | 0.006 | 0.007 | −0.011 | 0.010 | 0.009 |
| 25–50 | CH | 0.010 | 0.010 | 0.018 | 0.018 | 0.011 | 0.019 | 0.010 | 0.019 | 0.004 |
| | DCH | −0.009 | −0.007 | 0.007 | 0.003 | 0.006 | 0.007 | −0.009 | 0.007 | 0.007 |
| 50–75 | CH | 0.025 | 0.022 | 0.020 | 0.023 | 0.012 | 0.020 | 0.012 | 0.025 | 0.005 |
| | DCH | −0.008 | −0.007 | 0.004 | 0.003 | 0.005 | 0.006 | −0.008 | 0.006 | 0.006 |
| 75–100 | CH | 0.045 | 0.033 | 0.026 | 0.033 | 0.013 | 0.030 | 0.013 | 0.045 | 0.010 |
| | DCH | −0.006 | −0.006 | 0.001 | 0.000 | 0.002 | 0.002 | −0.006 | 0.002 | 0.004 |

Table 6 examines six sectors (the same as Table 3). As before, the SoC was divided into four sections, and each of the six sectors was examined separately. The analysis range and variance were calculated by taking into account all sectors. The results show a more uniform deformation above the 25% charging range. However, no uniform deformation was observed at higher SoC levels and ranges. On this basis, determining the battery charge level from the deformation data is difficult. During discharge, there is a very slight contraction, but the rate of change is within the margin of error. As a result, the resulting values should be measured at a lower resolution or not used at all. Table 7's values are based on Figure 7, part (d).

Table 7 depicts the deformation of the various SoC levels. In the case of the segmentation into three sectors, the values on the A and B sides are also combined and averaged, reducing the amount of variation even further. Determining when a charge or discharge occurs is difficult using the values determined here. To summarize, minimal variation was observed for the tested Panasonic 25 Ah prismatic NMC battery. Based on these

findings, a tactile inspection is not recommended for this type of battery. Further tests were conducted using the DIC technique at the highest resolution possible. Although it cannot be determined at the SoC level, monitoring the battery's SoH is crucial because it is a semi-hard battery. This method is expected to identify extremely worn batteries. Furthermore, it is essential to note that several measurements were performed on one type of prismatic battery, so the entire design family does not represent the results.

**Table 7.** Deformation of a Panasonic 25 Ah battery with prismatic NMC composition in the cases of 3 sectors.

| SOC [%] | CH/DCH | Sec_1 | Sec_2 | Sec_3 | Variance Range [mm] | | Deviation [mm] |
|---|---|---|---|---|---|---|---|
| 0–25 | CH | 0.006 | 0.009 | 0.008 | 0.006 | 0.009 | 0.002 |
| | DCH | −0.007 | 0.004 | 0.008 | −0.007 | 0.008 | 0.008 |
| 25–50 | CH | 0.010 | 0.018 | 0.015 | 0.010 | 0.018 | 0.004 |
| | DCH | −0.005 | 0.003 | 0.007 | −0.005 | 0.007 | 0.006 |
| 50–75 | CH | 0.018 | 0.020 | 0.016 | 0.016 | 0.020 | 0.002 |
| | DCH | −0.005 | 0.003 | 0.005 | −0.005 | 0.005 | 0.005 |
| 75–100 | CH | 0.022 | 0.026 | 0.020 | 0.020 | 0.026 | 0.003 |
| | DCH | −0.004 | 0.001 | 0.002 | −0.004 | 0.002 | 0.003 |

### 3.3. Pouch-Shaped Battery Analysis

Turnigy 5 Ah lithium polymer batteries were used as the pouch-type batteries. During the tests, both newer and worn-out batteries were measured. Digitalization at the time of discharge and a charge interruption were also performed during these tests. Figure 8 depicts the deformation of lithium polymer batteries.

The results of the DIC measurements are shown in Figure 8. The three states corresponding to the precise measurement during charging and discharging are highlighted in Figure 8, and several interruptions are reported in tables later. Each measurement interruption is accompanied by a data table containing side A and B deformation data, which are merged below. The evaluation procedure is depicted in Figure 9.

The analysis of the pouch cell's deformation data, where the two sides are already juxtaposed, can be seen in Figure 9. For the evaluation, a 1015 mesh was created, which was then reduced to 45 sectors during the reduction process. It is important to note that the data in this matrix mesh is not always distributed evenly; the average of the available values was always used. In addition, columns 1-5 contain the values for the cell's side A, and columns 6–10 contain the values for the cell's side B. Two approaches similar to the previous ones were used to process the data to determine the critical zone. In the first case, the cell is divided into two parts (Figure 9c), the middle and the upper. The researchers can then compare each zone to the others. The second method (Figure 9d) divides the cell into lower, middle, and upper sections. After that, the electrical and digitalization data were combined. Table 8 shows the averaged deformation values measured during the charging and discharging of several lithium polymer Turnigy 5 Ah batteries. The values in the table correspond to Figure 9, part (a).

The charge level values shown in the first row of Table 8 are in percentages, while the values in columns 3, 4, and 5 are in millimeters. Section 3.1 (based on Table 2) describes in detail how to process the values from Table 8. The deformation values are from the initial state and are compared to the first model in all cases, but each measurement has a new initial state. For each SoC level, five minimum (discharge) and five maximum (charge) values were used in the table and calculation. In this case, the results show that the changes are much more significant in decimal order. The analysis shows that the different states can be well-separated at different charge levels. During discharge, there is an obvious

constriction, and during charging, there is an obvious expansion. However, the range and variance are quite wide. Table 9 is based on Figure 9, section (c).

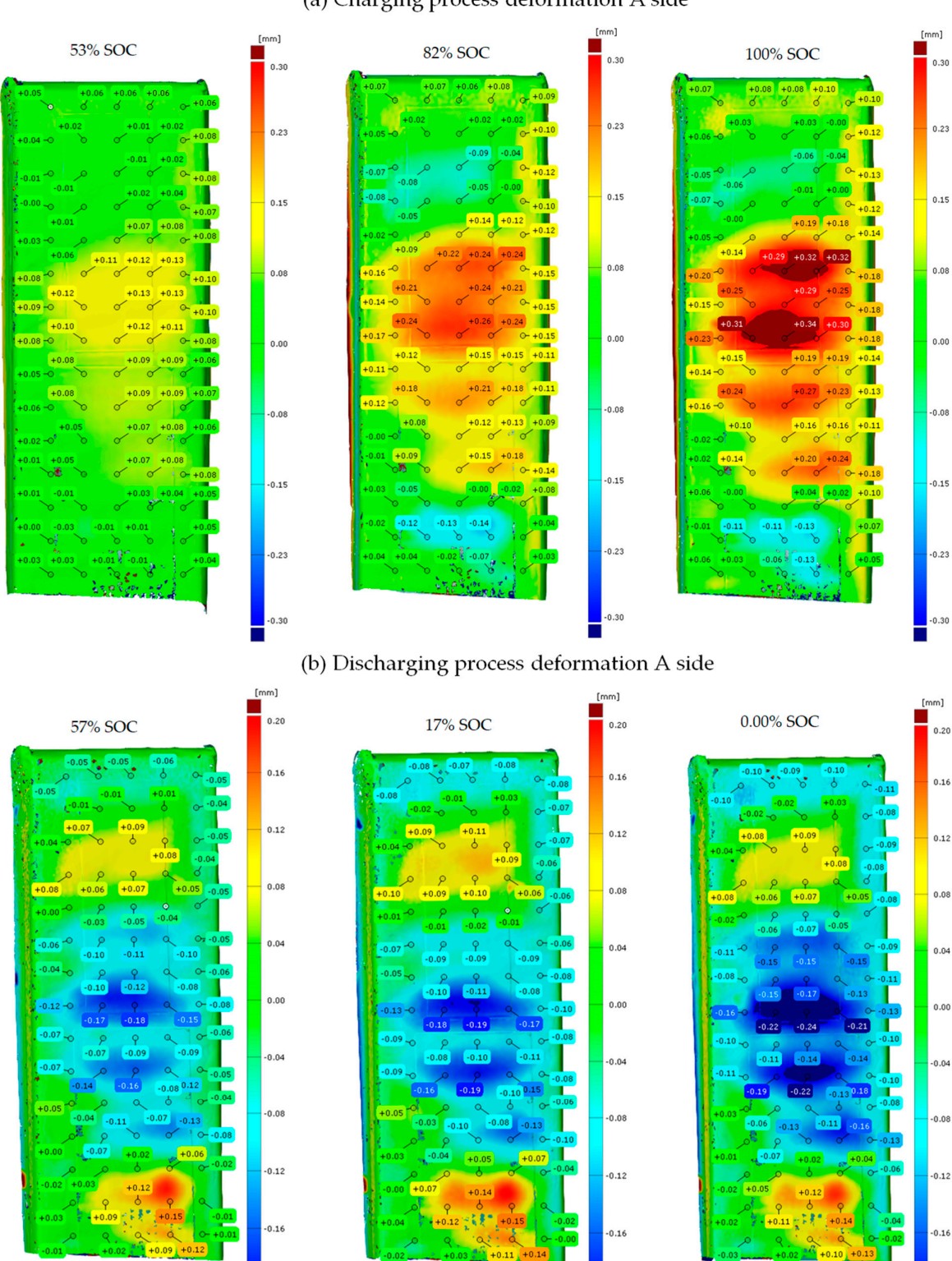

**Figure 8.** (**a**) Pouch-type battery deformation during charging; (**b**) Pouch-type battery deformation during discharge.

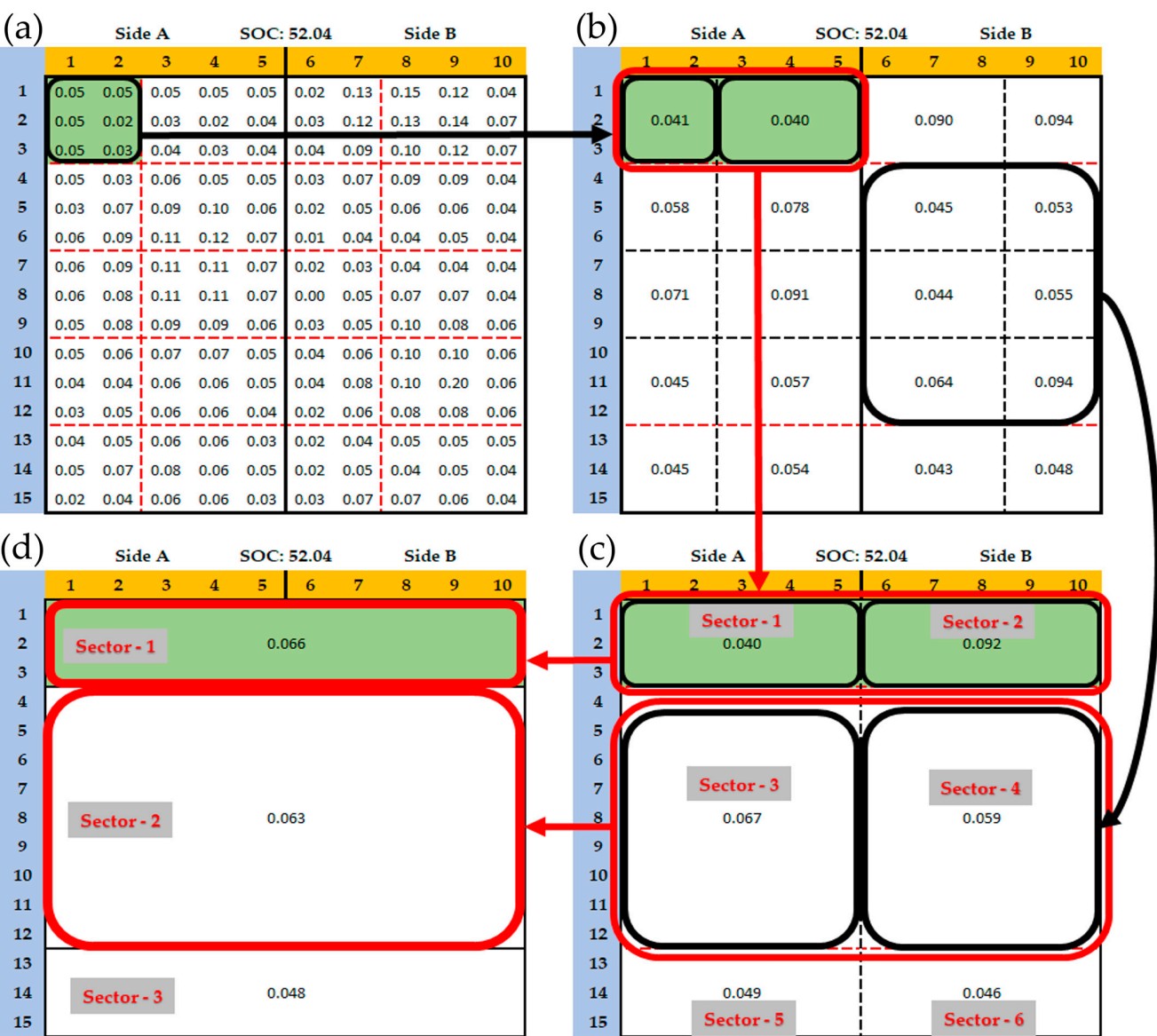

**Figure 9.** Assignment of critical zones in the pouch design lithium polymer battery: (**a**) Values extracted from the GOM file at a constant distance; (**b**) 6 × 3 sector separation; (**c**) 6-sector separation; (**d**) 3-sector separation.

**Table 8.** Deformation of the lithium polymer Turnigy 5 Ah battery based on min/max values measured over the entire surface.

| SOC [%] | CH/DCH | Deformation [mm] | Variance Range [mm] | | Deviation [mm] |
|---|---|---|---|---|---|
| 0–25 | CH | 0.089 | 0.036 | 0.127 | 0.047 |
| | DCH | −0.19 | −0.17 | −0.2 | 0.013 |
| 25–50 | CH | 0.143 | 0.104 | 0.171 | 0.035 |
| | DCH | −0.17 | −0.16 | −0.17 | 0.008 |
| 50–75 | CH | 0.192 | 0.139 | 0.23 | 0.047 |
| | DCH | −0.13 | −0.11 | −0.15 | 0.023 |
| 75–100 | CH | 0.235 | 0.225 | 0.25 | 0.013 |
| | DCH | −0.08 | −0.04 | −0.11 | 0.035 |

**Table 9.** Deformation of the pouch-designed lithium polymer Turnigy 5 Ah battery in the cases of 6 sectors.

| SOC [%] | CH/DCH | Sec_1 | Sec_2 | Sec_3 | Sec_4 | Sec_5 | Sec_6 | Variance Range [mm] | | Deviation [mm] |
|---|---|---|---|---|---|---|---|---|---|---|
| 0–25 | CH | 0.013 | 0.021 | 0.031 | 0.032 | 0.013 | 0.017 | 0.013 | 0.032 | 0.008 |
| | DCH | −0.069 | −0.058 | −0.064 | −0.060 | 0.011 | 0.024 | −0.069 | 0.024 | 0.042 |
| 25–50 | CH | 0.036 | 0.043 | 0.061 | 0.061 | 0.027 | 0.030 | 0.027 | 0.061 | 0.015 |
| | DCH | −0.043 | −0.035 | −0.049 | −0.023 | –0.016 | 0.017 | −0.049 | 0.017 | 0.024 |
| 50–75 | CH | 0.057 | 0.077 | 0.089 | 0.080 | 0.056 | 0.040 | 0.040 | 0.089 | 0.018 |
| | DCH | −0.043 | −0.025 | −0.039 | −0.045 | −0.014 | 0.015 | −0.045 | 0.015 | 0.023 |
| 75–100 | CH | 0.081 | 0.076 | 0.104 | 0.099 | 0.071 | 0.024 | 0.024 | 0.104 | 0.029 |
| | DCH | −0.019 | −0.009 | −0.020 | −0.028 | –0.004 | 0.017 | −0.028 | 0.017 | 0.016 |

In addition, six different sectors are examined in Table 9 (the same as in Table 3). Two sectors are analyzed for the upper part, two for the middle part, and two for the central part of the deformation analysis. It is worth noting that the two middle sectors are larger than the top and bottom sectors, and the cell's A and B sides are separated. As a result, three sectors can be seen from side A and three from side B. The test range and variance were calculated considering all sectors in this case. The most significant variation in deformation was found in sectors 3 and 4, but the variation was well-separable in most cases. The value variance is relatively large, especially when the discharge is less than 25% and the charging is greater than 75%. Table 10 is based on Figure 9, section (d).

**Table 10.** Deformation of the pouch-designed lithium polymer Turnigy 5 Ah battery in the cases of 3 sectors.

| SOC [%] | CH/DCH | Sec_1 | Sec_2 | Sec_3 | Variance Range [mm] | | Deviation [mm] |
|---|---|---|---|---|---|---|---|
| 0–25 | CH | 0.024 | 0.031 | 0.016 | 0.005 | 0.048 | 0.015 |
| | DCH | −0.063 | −0.061 | −0.006 | −0.075 | −0.003 | 0.029 |
| 25–50 | CH | 0.048 | 0.059 | 0.031 | 0.022 | 0.071 | 0.018 |
| | DCH | −0.052 | −0.055 | 0.000 | −0.061 | 0.003 | 0.027 |
| 50–75 | CH | 0.072 | 0.085 | 0.048 | 0.042 | 0.100 | 0.019 |
| | DCH | −0.034 | −0.042 | 0.000 | −0.050 | 0.011 | 0.021 |
| 75–100 | CH | 0.082 | 0.105 | 0.046 | 0.042 | 0.110 | 0.026 |
| | DCH | −0.014 | −0.024 | 0.007 | −0.033 | 0.011 | 0.015 |

The SoC in Table 10 is level; the other values are related to deformation and are in mm. The analysis procedure was the same as before (see Tables 4 and 7).

Average values were used appropriately in the sector transformations. It can be seen that the most significant change occurs during charging in the middle (2) sector. This is so severe that it can be detected in that sector using tactile sensing. Sectors 1 and 2 can already be used for this purpose during discharge.

In summary, the deformation of the pouch batteries studied in this study, including lithium polymer Turnigy 5 Ah cells, varied by tenths of an order of magnitude. Because of the rate of change, this type can be detected using instruments other than DIC with lower precision. A state discharged to 0 V was investigated in subsequent tests. It was discovered that such a deep discharge causes cell expansion rather than contraction. Based on these few deep discharges, contraction was found to be replaced by expansion in the range below 3 V. This could be an essential area to investigate in future battery diagnostics.

## 4. Discussion

The measurements taken will be used to develop an intelligent battery testing procedure that will consider variables other than capacity tests and electrical parameter assessment. As the number of batteries in use grows, testing time becomes increasingly important. On the one hand, developing the most rapid and accurate diagnostic procedure possible is critical. On the other hand, classifying and grouping batteries of various discharges, types, and conditions based on their degree of wear is difficult. A test that considers battery deformation can increase the speed and precision of cell tests. When combined with a deformation measurement, the DIC measurement method provides high accuracy. The results' applicability is in the field of the condition assessment of automotive and commercially used batteries, which was our goal in the first place. From an automotive standpoint, promoting reuse is critical, which can be accomplished by taking the following steps:

- Detection and localization of faulty cells during in-vehicle use [54].
- Removal of identified defective battery packs.
- Dismantling of battery packs, and acceptance of worn cells.
- Initiate inspection procedures, carrying out basic safety measurements (short circuit, damage tests).
- Optical inspection by surface digitization and classification of cells that can be clearly identified. The more measurement data entered into the system, the more accurate it will be. This is the fastest way of testing, as no physical connection to the cell is required. Easy to automate.
- Optical testing with battery testing. In this case, the cells are already loaded and charged with special test signals. This procedure is also more likely to be able to provide a status qualification at the beginning.
    ○ Initially, using complete surface digitization, testing at different SOC levels. The evaluation is compared to the initial state.
    ○ Subsequently, up to a few images and at a few SOC levels. This requires more information on the cell state.
- Displacement (tactile) sensors and battery testing. In the next phase, the cells would be tested in individual measuring frames or by robots. Here, after loading, the deformation of the cells would be tested with special load/charge signals using distance sensors (if possible).
- The last case is the implementation of distance sensors and the full-capacity test. A full-capacity test is required if the first three quick measurements do not give satisfactory results. This is a long run but gives a high-accuracy result.

If the cell status can be determined clearly at any point, the battery is considered "tested" (no further action is required), and a new battery to be tested is added to the system. Connecting artificial intelligence to the system and the database would help improve testing speed by allowing for more measurements. In other words, the more measurement results that are available, the faster the scanning and selection will be.

Although the temperature of the cells to be tested is critical, climate chamber measurements are required to accurately establish the relationship between the deformation and temperature effects. Measurements are made more difficult because not only the battery but also the DIC measuring system must be inserted before measurements can be taken. Furthermore, possible high-temperature values may cause additional inaccuracies [55,56]. Another issue is calibration; optical measuring instruments must be recalibrated for each temperature step because they are very sensitive to temperature changes. The accuracy and reliability of strain and displacement measurements are greatly influenced by the calibration of a digital image correlation (DIC) system. Calibration is critical for minimizing errors introduced by different variables in the measurement chain [57–59]. As a result, the method developed and presented here can currently be used at a temperature of 25 °C +/− 5 °C. The results may differ from those obtained using other temperature scales.

The selection of load is another critical aspect of the tests. This topic has already been covered in another publication [60]. Based on the findings, the rate of deforestation remained constant, resulting in the same maximum change. The higher the load, however, the faster the run-down times (hump and contraction occur). These results were obtained in a pouch LiPo test; other cell types may have different results, but this is also where the most significant change was observed. Changing the C-rate speeds up the test, but 1C was used first for safety reasons. The test results are from a 1C load and charge and do not cover specific test signals (transient, multi-sine, wltp, and so on). We intend to investigate such cases in the future.

## 5. Conclusions

The study aims to create a complex battery condition assessment and diagnostic method to evaluate cell conditions using electrical and deformation data. It also determines the expected maximum deformation of various cell shapes and their locations and causes. The tests look at various battery designs. Cylindrical, prismatic, and pouch cells are the most commonly used in the tests. Batteries of various types were also tested in various States of Health. The measurements, however, did not include several representatives of each design. The results for the cylindrical cells presented are based on the 18650 size but different brands (Samsung, Panasonic, Sony, and so on). Several 25 Ah Pansonic cells with an NMC composition were tested in the Prizamtic case, and several 5 Ah Turnigy lithium polymer cells were tested for the pouch cells. In both cases, the measurement results revealed swelling during charging and contraction during discharging. The extent of this depended heavily on the design. The results of the analyses were presented in two ways, with one based on the change in the total surface area of the battery. The other option was to build different sectors. The results show that the deformation rate for 18650-size prismatic cells is much lower than for soft-case lithium polymer batteries. As a result, it is advised to identify the distinct outliers for the 18650 size and prismatic. Furthermore, the higher the aggregation, the more difficult it is to predict the SoC levels for critical zone determination. Both methods work well in lithium polymer pouch cells, and sectors can be assigned. Furthermore, the magnitude of the variation for the lithium polymer batteries studied was in the tenths, whereas it was in the hundreds for the cylindrical and prismatic cases. Thus, at this stage of research, it is possible to conclude that the relationship between the SoC level estimation and deformation in the case of the lithium polymer warrants further investigation. The diagnosis based on DIC images is expected to yield better results in prismatic and cylindrical cases (SoH estimation).

**Author Contributions:** Conceptualization, S.K.S., M.S., S.S. and S.F.; methodology, S.K.S., M.S., S.S. and S.F.; software, S.K.S., M.S., S.S. and S.F.; validation, S.K.S., M.S., S.S. and S.F.; formal analysis, S.K.S., M.S., S.S. and S.F.; investigation, S.K.S., M.S., S.S. and S.F.; resources, S.K.S., M.S., S.S. and S.F.; data curation, S.K.S., M.S., S.S. and S.F.; writing—original draft preparation, S.K.S., M.S., S.S. and S.F.; writing—review and editing, S.K.S., M.S., S.S. and S.F.; visualization, S.K.S., M.S., S.S. and S.F.; supervision, S.K.S., M.S., S.S. and S.F.; project administration, S.K.S., M.S., S.S. and S.F.; funding acquisition, S.K.S., M.S., S.S. and S.F. All authors have read and agreed to the published version of the manuscript.

**Funding:** This research received no external funding.

**Data Availability Statement:** Data are contained within the article.

**Acknowledgments:** This paper was technically supported by the research team entitled "SZE-RAIL".

**Conflicts of Interest:** The authors declare no conflicts of interest.

## List of Abbreviations

| | |
|---|---|
| AE | Acoustic Emission |
| ANN | Artificial Neural Networks |
| BMS | Battery Management Systems |
| BTM | Battery Thermal Management |
| CH | Battery Charging Process |
| CNN | Convolutional Neural Networks |
| CT | Computer Tomography |
| DCH | Battery Discharging process |
| DIC | Digital Image Correlation |
| DoD | Depth of Discharge |
| EIS | Electrochemical Impedance Spectroscopy |
| LDs | Lithium Dendrites |
| Li-ion | Lithium-ion Battery |
| LiPo | Lithium Polymer Battery |
| LSTM | Long Short-Term Memory |
| LFP | Lithium Ferrophosphate |
| ML | Machine Learning |
| NMC | Lithium Nickel Manganese Cobalt Oxide |
| SEI | Solid Electrolyte Interphase |
| SEM | Scanning Electron Microscope |
| SoC | State of Charge |
| SoH | State of Health |
| TR | Thermal Runaway |

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
