# Peer review of "Deformation Analysis of Different Lithium Battery Designs Using the DIC Technique"

_energies, doi:10.3390/en17020323_

Round 1

Reviewer 1 Report

Comments and Suggestions for Authors

Comments

Actually, the failure of battery systems is caused by a few composed cells, which results in the breakdown of the whole battery system. Therefore, if we can find the damaged cells, we can repair or replace them to extend the use life of the battery system.

It is reasonable to recognize the deformation of the battery system during the charging and discarded process. The authors aim to develop a complex battery condition assessment and diagnostic method that can assess cell conditions based on electrical and deformation data. The digital image correlation (DIC) technique can be used for cell state estimation. The tests were conducted with several cells of commonly used geometries and encapsulations, such as lithium polymer (soft casing), 18650 standard sizes (hard casing), and prismatic cells (semi-hard). Based on different materials and hardness, it is easy to understand that the deformation of soft case lithium polymer batteries is much higher than that of 18650 size and prismatic cells. The research is very important for the further use of lithium-ion battery systems. However, there are some problems in the manuscript, and the manuscript should be accepted after major revision and well organized. The detailed comments are as follows:

1. Even though the (DIC) technique can test the deformation of the battery system, where will it be equipped? The automobile, recycling workshop, or battery repair workshop?

2. This high accuracy of the test will bring high cost, is it economical to repair the damaged battery system by the method, and how to repair the damaged battery system?

3. If there are some open circuits in the damaged battery (charging and discharging are not allowed), how to test its deformation?

4. Which is economical and effective to achieve high capacities, high cycle life, and safety for battery systems? To repair the damaged battery system, or to recycle the essential elements to encapsulate a new battery system?

Comments on the Quality of English Language

Minor editing of English language required

Author Response

Dear Reviewer,

many thanks for the valuable comments. You can find our answers point-by-point in the attached document. At the end of the PDF, there is a comparison section where all the changes have been tracked; you can check what we modified in the revised manuscript compared to the original submitted one.

We hope you will be satisfied with the improvements.

Yours sincerely,

The Authors

Reviewer 2 Report

Comments and Suggestions for Authors

The present work is novel and interesting, however, major modifications are required.

1-      The introduction need to highlight the impact and problems of various commercial cathode materials like LFP, NMC, ..etc on the performance of each cell design. Hence, some relevant and recent studies should be cited including Colloids and Surfaces A: Physicochemical and Engineering Aspects 672 (2023) 131748.

2-      The digital diagnostic imaging of te batteries under charge discharge conditions should be done and compared at each cell design at different C-rates.

3-      The diagnostic study should consider the imaging at elevated temperatures.

Comments on the Quality of English Language

The whole paper English need to be improved

Author Response

Dear Reviewer,

many thanks for the valuable comments. You can find our answers point-by-point in the attached document. At the end of the PDF, there is a comparison section where all the changes have been tracked; you can check what we modified in the revised manuscript compared to the original submitted one.

We hope that you will be satisfied with the improvements.

Yours sincerely,

The Authors

Reviewer 3 Report

Comments and Suggestions for Authors

This research proposed a complex battery condition assessment and diagnostic method regarding to electrical and deformation data. The method and results are clearly demonstrated. It is recommended to accept after minor revision.

1. Please define all acronyms at their first mention and better put them into a table.

2. The novelty of the current study can be further clarified by considering more battery-related reviews. i.e., 3.

1) Liu, Y., Wang, L., Li, D., & Wang, K. (2023). State-of-health estimation of lithium-ion batteries based on electrochemical impedance spectroscopy: a review. Protection and Control of Modern Power Systems8(1), 41.

2) Li, A., Weng, J., Yuen, A. C. Y., Wang, W., Liu, H., Lee, E. W. M., ... & Yeoh, G. H. (2023). Machine learning assisted advanced battery thermal management system: A state-of-the-art review. Journal of Energy Storage60, 106688.

3) Chavan, S., Venkateswarlu, B., Prabakaran, R., Salman, M., Joo, S. W., Choi, G. S., & Kim, S. C. (2023). Thermal runaway and mitigation strategies for electric vehicle lithium-ion batteries using battery cooling approach: A review of the current status and challenges. Journal of Energy Storage72, 108569.

3. The potential application of the proposed method and the enlightenment of current results should be further elaborated.

Author Response

(The authors gave the same response as above.)

Round 2

Reviewer 1 Report

Comments and Suggestions for Authors

The revised manuscript can be accepted after minor English corrections.

Comments on the Quality of English Language

Minor editing of the English language is required

Author Response

Dear Editors, Dear Reviewers,

we are very grateful that you prepared the reviews of our article submitted in Energies (MDPI). We answered all of your comments and questions point-by-point. We trust that we have not missed anything and that the responses are satisfactory. We are glad that we were able to improve our manuscript based on your comments and guidance. We hope that you will be satisfied with the modified, revised form of the paper, and based on them, you can recommend it for publishing. Finally, we appreciate your effort and taking the time to prepare these detailed and helpful reviews.
At the end of the attached PDF document, you can find a comparison section where all the changes have been tracked. You can check all of the modifications we applied in the revised version.

Yours sincerely,

The Authors

Reviewer 2 Report

Comments and Suggestions for Authors

The novelty of the work still not clear and the significance of the work is not present. More statement and clarification for the novelty should be added to the introduction.  Also, the impact and problems of various commercial cathode materials like LFP, NMC, .etc on the performance of each cell design should be added. Hence, some relevant and recent studies should be cited including Colloids and Surfaces A: Physicochemical and Engineering Aspects 672 (2023) 131748.

Comments on the Quality of English Language

The English of the whole paper need to be improved

Author Response

(The authors gave the same response as above.)

Round 3

Reviewer 2 Report

Comments and Suggestions for Authors

The comments are carefully answered and the paper is now accepted

Comments on the Quality of English Language

Accepted